# Physicochemical Properties and Biological Characteristics of *Sargassum fusiforme* Polysaccharides Prepared through Fermentation of *Lactobacillus*

Ying Yang [1,2], Dan Ouyang [1,2], Jiayao Song [1,2], Chunyang Chen [1,2], Chenjing Yin [1,2], Laijin Su [1,2,*] and Mingjiang Wu [1,2,*]

1   College of Life and Environmental Science, Wenzhou University, Wenzhou 325035, China; yy00116@163.com (Y.Y.); ou_r@foxmail.com (D.O.); sjy152572@163.com (J.S.); 18842857579@163.com (C.C.); 18850637059@163.com (C.Y.)
2   Zhejiang Provincial Key Laboratory for Water Environment and Marine Biological Resources Protection, Wenzhou University, Wenzhou 325035, China
*   Correspondence: sulj@wzu.edu.cn (L.S.); wmj@wzu.edu.cn (M.W.)

**Abstract:** *Sargassum fusiforme* polysaccharides (SFPs) have multiple activities. The fermentation of *S. fusiforme* by *Lactobacillus* can alter its polysaccharide properties and biological activities. In this study, three different *Lactobacillus* species (*Lactobacillus plantarum* (LP), *Lactobacillus acidophilus* (LA), and *Lactobacillus rhamnosus* (LR)) were selected to ferment *S. fusiforme*. The polysaccharides SFP (unfermented) and FSFP (fermented by LP, LA, or LR denoted as LP-SFP, LA-SFP, and LR-SFP, respectively) were extracted, and their physicochemical properties and biological activities were investigated. According to the results, fermentation caused significant changes in the physicochemical properties and biological activities of SFP. Specifically, FSFP showed a significant increase in uronic acid and fucose content and a significant decrease in molecular weight; LA-SFP and LR-SFP had stronger DPPH scavenging abilities; LR-SFP had the strongest inhibition of ROS production and cell mortality; LP-SFP and LR-SFP significantly increased SOD activity in zebrafish; LA-SFP had a significant effect on the proliferation of *Lactobacillus plantarum*; LP-SFP had a significant effect on the proliferation of *Lactobacillus rhamnosus*; and LA-SFP had a stronger food-excretion-promoting activity. In conclusion, the fermentation of *Lactobacillus* for the preparation of SFPs can change the physicochemical properties of polysaccharides and has broad potential for improving their biological activity.

**Keywords:** *Sargassum fusiforme* polysaccharides; fermentation; *Lactobacillus*; biological activity

## 1. Introduction

*Sargassum fusiforme* belongs to the family Sargassaceae and grows mainly in the coastal areas of southeastern China, Japan, Korea, and South Korea [1,2]. *S. fusiforme* polysaccharides (SFPs) are the main component of *S. fusiforme* and have biological activities such as antitumor activity, antioxidant activity, immunity enhancement, hypoglycemia, hypolipidemia, and the regulation of intestinal flora [3,4]. SFPs primarily consist of fucoidan, alginate, and laminaran. Most of the nutrients in *S. fusiforme* are water-soluble and can be extracted using hot water or acid; however, water extraction is time-consuming and has a low yield, while acid extraction tends to degrade and destroy the composition and structure of polysaccharides that are crucial to its activity [5,6]. Biomodification has received increasing attention because of its simplicity, efficiency, and environmental friendliness [7,8]. Enzymatic hydrolysis and fermentation in biological modifications have been shown to change the cell wall structure and promote the release of soluble components, which can increase cell wall permeability, promote the release of active ingredients, and improve their functional activity [9].

Lactic acid bacteria (LAB) are commonly used as probiotic fermenters. LAB fermentation is a traditional, safe, and economical biotechnology process used for the production of functional foods [10,11], which can modify the characteristics and types of bioactive compounds [12] and affect their biological activity. *Lactobacillus plantarum* (LP), *Lactobacillus rhamnosus* (LR), and *Lactobacillus acidophilus* (LA) are widely used in the food industry, and their probiotic function in human health is generally accepted, suggesting that their proliferation can be used to assess the prebiotic activity of compounds. In addition to their role as probiotics, they also produce healthy fermented foods. Probiotic–diet interactions are currently popular topics in probiotic science research [13,14]. Dietary fiber generally refers to carbohydrates that cannot be absorbed and metabolized by the small intestine but can be partially fermented by intestinal flora in the large intestine and was once mistakenly considered non-nutritious and not valued [15]. In recent years, a large number of studies have reported that probiotics have specific enzyme systems that can transform dietary components through fermentation, promote the dissolution rate of bioactive substances to be significantly increased, and reduce molecular weight so that they are more easily absorbed and their activity is also increased [14]. Therefore, the advantages of probiotic fermentation are gradually recognized, and research is hot in the field of plant polysaccharide preparation.

In this study, fermented SFPs (FSFP: LP-SFP, LA-SFP, and LR-SFP) were isolated from the fermentation broth using *S. fusiforme* as the raw material. The physicochemical properties, chemical structure, and in vitro and in vivo biological activities of the polysaccharides were studied and compared with those of unfermented SFP (water extracted). LABs for the fermentation of SFP were screened to provide a theoretical basis and technical reference for the preparation of highly active SFP by microbial modification.

## 2. Materials and Methods

### 2.1. Materials and Reagents

*S. fusiforme* was collected from Dongtou District (Wenzhou, Zhejiang, China). Dextran standard was purchased from the National Institute for Food and Drug Control (Beijing, China). D-Glucose, D-Galactose, L-Arabinose, L-Rhamnose, D-Xylose, D-Mannose, L-Fucose, D-Galacturonic acid, and D-Glucuronide were purchased from Yuanye (Shanghai, China). Oligofructose was purchased from Macklin (Shanghai, China). All of the other chemicals were of analytical grade.

### 2.2. Bacterial Strains

The bacterial strains included *Lactobacillus plantarum* (GDMCC 1.380), *Lactobacillus acidophilus* (GDMCC 1.731), and *Lactobacillus rhamnosus* (GDMCC 1.325), which were purchased from the Guangdong Microbial Culture Collection Center (Guangzhou, China). The strains were stored in MRS broth containing 20% glycerol at −80 °C for later use. The strains were activated before being used for *S. fusiforme* fermentation [16].

### 2.3. Preparation of Fermented SFPs

*S. fusiforme* was washed, dried, and crushed. Before fermentation, 30 g of algae powder was taken, water was added in a material–liquid ratio of 1:10, and the pH was adjusted to 6.8 ± 0.1, followed by sterilization for 20 min at 121 °C. Bacterial broth containing 4% (*v*/*v*) LP, LA, or LR was added to a final concentration of 6.5 log CFU/mL, and static fermentation was carried out at 37 °C for 24 h. A sterile water control was used instead of bacterial broth. The fermentation reaction for both the fermented (FSFP) and unfermented (SFP) groups was stopped by heating at 100 °C for 10 min, the supernatant was collected by centrifugation, and the final pH of the fermentation broth was determined to be 4.7 ± 0.1. The supernatant was concentrated under reduced pressure, and 3 times the volume of anhydrous ethanol was added. The resulting precipitate was obtained by centrifugation at 4 °C for 10 h. Fermentation of *S. fusiforme* with LR, LA, and LR resulted in fermented SFP

(FSFP: LP-SFP, LA-SFP, and LR-SFP, respectively), while unfermented SFP was obtained through water extraction.

*2.4. Physicochemical Characteristics of SFPs*

2.4.1. Chemical Characteristics

The total sugar content was determined by using the phenol-sulfuric acid method [17]; protein concentration was determined by using the Bradford assay [18]; the content of uronic acid was determined by using the m-hydroxybiphenyl method [19]; the sulfate content was determined by using the $BaCl_2$ gelatin turbidimetric method [20]; and the content of fucoidan was determined using the hypo-methyl blue turbidimetric method [21]. The content of alginate was determined according to the method of Shang et al. [22]. In short, the instructions are as follows: accurately weigh 1 g of the sample, add 2 mol/L hydrochloric acid 30 mL to soak for 12 h, filter, wash the filter residue of the filtrate, add silver nitrate solution to the filtrate without white precipitation, mix filter residue with 30 mL 0.1 mol/L calcium acetate solution, soak for 2 h, add 50 mL double-distilled water to mix evenly, use phenolphthalein as an indicator, and titrate with sodium hydroxide standard solution. Alginate content was calculated according to the following formula:

$$X\,(\%) = (((V - V0) \times C \times 0.2160)/m) \times 100 \qquad (1)$$

where V is the volume of sodium hydroxide standard solution consumed by the titration sample (mL), V0 is the volume of sodium hydroxide standard solution consumed by the titration blank sample (mL), C is the concentration of sodium hydroxide standard solution (mol/L), 0.2160 is the mass of sodium alginate comparable to 1.00 mL of sodium hydroxide standard titration solution [c(NaOH) = 1.000 mol/L] (g), m is the sample mass (g), and 100 is the unit conversion factor.

The monosaccharide composition of FSFP and SFP was determined using the PMP (1-phenyl-3-methyl-5-pyrazolinone) derivatization method [23]. A Waters XBridge C18 (4.6 mm × 250 mm, 5 μm) column was used with a 1260 Infinity II high-performance liquid chromatography (HPLC) system (Agilent, USA) equipped with a DAD detector. Mobile phase A was phosphate buffer solution ($Na_2HPO_4$-$NaH_2PO_4$) (PBS 0.1 mol/L pH = 6.9), mobile phase B was acetonitrile, phase A:B = 83:17, the flow rate was 1 mL/min, the injection volume was 20 μL, detection wavelength was 254 nm, and the column temperature was 25 °C.

2.4.2. Molecular Weight (MW) Analysis

The molecular weight of the polysaccharides was determined using high-performance size exclusion chromatography (HPSEC) [24]. A TSK-gel G-5000 PWXL column (7.5 mm × 300 mm, 10 μm) was used with a 1260 Infinity II HPLC system (Agilent) equipped with a 1260 RID differential detector. Dextran was used as the molecular weight standard (180, 2700, 9750, 36,800, 135,350, 300,600, and 2,000,000 Da), 0.2 mmol/L $Na_2SO_4$ solution was eluted at a flow rate of 0.5 mL/min, and the polysaccharide concentration was 3 mg/mL with a sample volume of 20 μL. The molecular weights of the samples were calculated according to the standard curve and elution volume (Ve) of the samples.

2.4.3. Fourier Transform Infrared (FT-IR) Spectroscopy

The dried samples were taken, pressed with KBr, and scanned using FTIR in the range of 4000–400 $cm^{-1}$, and the infrared spectra were recorded [25].

*2.5. Antioxidant Activity Analysis*

2.5.1. DPPH Free Radical Scavenging Rate Measurement

The method was described by Bursal et al. [26]. For the DPPH assay, 100 μL of DPPH ethanol solution was mixed with 100 μL of polysaccharide solution (0.50, 1.00, 1.50, 2.00, 1.50, and 3.00 mg/mL) in a 96-well plate. The reaction was carried out for 30 min, protected

from light, and the absorbance was measured at 517 nm. The DPPH radical-scavenging rate was calculated using the following equation:

$$\text{DPPH radical scavenging activity (\%)} = (1 - (A1 - A2)/A0) \times 100 \tag{2}$$

where A1 is the absorbance of the polysaccharide solution reacting with DPPH, A2 is the absorbance of the mixture of ethanol and polysaccharide, and A0 is the absorbance of the mixture of water and DPPH.

### 2.5.2. ABTS Free Radical Scavenging Rate Measurement

The method was described by Bursal et al. [26]. ABTS solution (100 μL) was mixed with 100 μL polysaccharide solution (0.1, 0.2, 0.3, 0.4, and 0.5 mg/mL) in a 96-well plate, protected from light for 10 min, and absorbance was measured at 734 nm. The ABTS radical-scavenging rate was calculated using the following equation:

$$\text{ABTS radical scavenging activity (\%)} = (1 - (A1 - A2)/A0) \times 100 \tag{3}$$

where A1 is the absorbance of the polysaccharide solution reacting with ABTS, A2 is the absorbance of the mixture of ethanol and polysaccharide, and A0 is the absorbance of the mixture of water and ABTS.

### 2.6. Prebiotic Activity Analysis

The method was as described by Wang et al. [27]. Two *Lactobacillus* species (LP and LA) were used to study the in vitro prebiotic activities of FSFP and SFP. The preparation of the medium was as follows: (1) basal medium: 0.05% (*w/v*) L-cysteine in carbohydrate-free MRS broth, and (2) proliferation medium: basal medium + FOS/polysaccharide, with FOS and polysaccharide (both filter-sterilized) at a final concentration of 2% (*w/v*). The basal medium was used as a blank control and the FOS proliferation medium as a positive control. Activated 10% (*v/v*) LP and LA were added to the medium and then incubated at 37 °C for 48 h. After 0 and 48 h of incubation, 1 mL of the culture solution was serially diluted with sterile saline, and then 100 μL of the diluted solution was spread on an MRS solid plate medium, with three parallels for each concentration, and incubated anaerobically at 37 °C for 48 h. The number of viable bacteria per milliliter of culture solution was determined by using the plate colony counting method. The proliferation of probiotic bacteria was determined by calculating the difference between the number of viable bacteria at 48 and 0 h in the culture solution using the following formula, and the results were expressed as log CFU/mL.

$$\text{Increase in bacterial number (log CFU/mL)} = \log B - \log A \tag{4}$$

where A is the number of viable bacteria at 0 h of incubation (CFU/mL) and B is the number of viable bacteria after 48 h of incubation (CFU/mL).

### 2.7. Antioxidant Activity Study Based on Zebrafish Model
### 2.7.1. Maintenance of Zebrafish

Zebrafish (*Danio rerio*) were purchased from Hunter Biotech (Hangzhou, China), and the maintenance conditions were described by Kim et al. [28]. Two males were bred with one female to obtain the embryos, which were harvested the next morning.

### 2.7.2. Polysaccharide and AAPH Treated Embryos

Eight hours after fertilization, embryos (*n* = 15) were transferred to 12-well plates containing E3 medium. The embryos were treated with a polysaccharide solution for 1 h and then with 15 mM AAPH for 24 h. Finally, the embryos were cultured to 3 dpf in E3 medium. The group without polysaccharide treatment served as a model control [29].

### 2.7.3. Estimation of AAPH-Induced Intracellular Reactive Oxygen Species (ROS) Production Rate and Cell Mortality

The ROS production rate and cell mortality were determined using 2,7-Dichloro fluorescein diacetate (DCFH-DA) (20 µg/mL) and acridine orange (7 µg/mL), respectively [30]. Zebrafish embryos were stained for 1 h in the absence of light, washed several times with embryo culture solution, and placed under a stereoscopic fluorescence microscope (Leica, Germany) to capture fluorescence photographs. Their fluorescence intensity was measured using ImageJ software, and the relative fluorescence intensity of each treatment group was calculated using the control group as the standard.

$$\text{Relative ROS production/cell death rate/lipid peroxidation rate (\%)} = \text{fluorescence intensity of the treatment group/fluorescence intensity of the control group} \times 100 \qquad (5)$$

### 2.7.4. Antioxidant Enzyme Activity Assay

Zebrafish embryos were collected and added to PBS to prepare a 10% tissue homogenate, and the supernatant was centrifuged and used for the determination of antioxidant-related indices. Superoxide dismutase (SOD) and catalase (CAT) activities were measured using a commercial kit (Nanjing Jiancheng Biological Engineering Research Institute).

### 2.8. Study on the Laxative Function Activity Based on Zebrafish Model

Zebrafish larvae with normal development five days after fertilization were transferred into 6-well plates, and 30 fish were placed in each well. The culture solution was removed from the plates, and 3 mL of Nile Red (10 ng/mL) was added and incubated for 16 h in the dark. The experimental groups were set up as follows: model control (zebrafish cultured with E3 culture solution after feeding with Nile Red) and polysaccharide-treated groups (zebrafish treated with polysaccharide solution after feeding with Nile Red). The subjects were treated with Nile Red for 24 h in the absence of light [31,32]. They were washed several times with the embryo culture solution and placed under a stereoscopic fluorescence microscope (Leica, Germany) to capture fluorescence photographs, and their fluorescence intensity was measured using ImageJ software.

$$\text{Food excretion promotion rate} = (1 - \text{fluorescence intensity of polysaccharide-treated group/fluorescence intensity of model control group}) \times 100\% \qquad (6)$$

### 2.9. Statistical Analysis

All experiments were repeated three times, and the data are expressed as mean ± standard error (SE). All statistical analyses were performed using GraphPad software (GraphPad Prism, version 8.0) for one-way analysis of variance to compare differences between groups, and $p < 0.05$ was considered statistically significant.

## 3. Results

### 3.1. Preliminary Characterization of SFPs

As shown in Table 1, chemical composition analysis revealed that SFP and FSFP mainly contained soluble sugars, uronic acid, alginate, and fucoidan. LA-SFP and LR-SFP contained significantly lower total sugars than SFPs, and FSFP had significantly lower alginate content, indicating that LA, LP, and LR may use carbohydrates leached from *S. fusiforme* as a carbon source to meet their own needs for normal growth and reproduction. He et al. [33] used *Lactobacillus fermentum* to ferment lychee and found that the polysaccharide content of lychee decreased after fermentation, which is similar to the results of this study. Compared to SFP, protein contents in LA-SFP and LR-SFP were significantly increased. This was likely due to the fermentation process. The extracellular enzymes produced by LA and LR disrupted the cell wall structure and released more protein. Compared to SFP, LR-SFP sulfate content was significantly increased, and LP-SFP,

LA-SFP, LR-SFP uronic acid content, and fucoidan content were significantly increased. The molecular weights of LP-SFP and LA-SFP were significantly reduced after fermentation.

**Table 1.** Physicochemical properties of SFP and FSFP.

| Sample | Total Sugar (%) | Sulphate (%) | Uronic Acid (%) | Alginate (%) | Fucoidan (%) | Protein (%) | MW (KDa) |
|---|---|---|---|---|---|---|---|
| SFP | 21.23 ± 0.35 [a] | 5.61 ± 0.06 [bc] | 13.41 ± 0.22 [c] | 22.69 ± 0.14 [a] | 11.11 ± 0.29 [d] | 0.31 ± 0.08 [b] | 27.21 ± 0.03 [a] |
| LP-SFP | 21.04 ± 0.48 [a] | 5.79 ± 0.28 [b] | 16.07 ± 1.27 [b] | 18.22 ± 0.15 [c] | 12.73 ± 0.60 [c] | 0.23 ± 0.16 [b] | 26.64 ± 0.11 [c] |
| LA-SFP | 14.31 ± 2.66 [c] | 5.45 ± 0.06 [c] | 15.90 ± 0.54 [b] | 18.83 ± 0.10 [b] | 16.92 ± 0.44 [b] | 0.93 ± 0.07 [a] | 26.86 ± 0.09 [b] |
| LR-SFP | 17.78 ± 1.65 [b] | 6.75 ± 0.16 [a] | 22.14 ± 0.88 [a] | 18.80 ± 0.08 [b] | 19.02 ± 0.44 [a] | 0.79 ± 0.04 [a] | 27.08 ± 0.04 [a] |

The use of the same letter indicates that there is no significant difference between the groups, while different letters indicate significant differences ($p < 0.05$).

The monosaccharide composition of the polysaccharide samples was determined by comparing the peak time with the chromatograms of the nine monosaccharide standards (Figure 1). The molar ratio of each monosaccharide was then calculated by combining the peak areas of each monosaccharide in the samples. As shown in Figure 1, fermentation with different LABs did not change the monosaccharide composition of SFP, which mainly consisted of seven monosaccharides, among which fucose and galactose were the most abundant. As shown in Table 2, fermentation, however, changed the monosaccharide ratio of the SFP, and the monosaccharide ratios of the polysaccharides prepared through different *Lactobacillus* fermentations differed. Unfermented SFP had the lowest proportion of glucuronic acid, LP-SFP had the highest proportion of rhamnose, and LR-SFP had the highest proportions of mannose, glucuronic acid, glucose, galactose, xylose, and fucose.

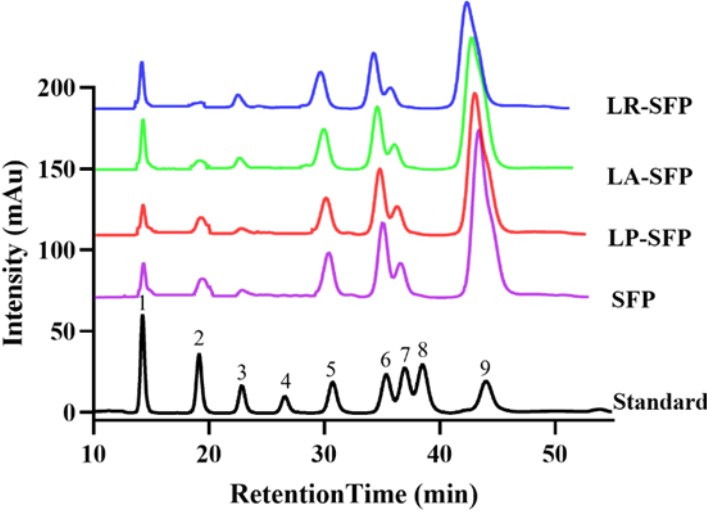

**Figure 1.** Liquid chromatogram of monosaccharide composition. (1-Mannose, 2-Rhamnose, 3-Glucuronic acid, 4-Galacturonic acid, 5-Glucose, 6-Galactose, 7-Xylose, 8-Arabinose, and 9-Fucose).

The FT-IR spectra of all samples are shown in Figure 2, and the spectra of all polysaccharide samples were similar. The band at 3423 cm$^{-1}$ has a very broad -OH stretching vibration absorption peak, and that at 2929 cm$^{-1}$ has a faint C-H bending and stretching vibration peak, which is characteristic of polysaccharides [34]. The presence of uronic acid was confirmed by the asymmetric stretching vibration of the carbonyl group at 1616 cm$^{-1}$ and the symmetric stretching vibration of -COOH at 1416 cm$^{-1}$. The absorption peak at 1041 cm$^{-1}$ corresponded to the stretching vibration of the monosaccharide ring and the C-O-C of the glycosidic bond [35]. The presence of sulfate groups, characteristic components of fucoidans, was indicated by the S = O stretching vibration peak at 1250 cm$^{-1}$ [36,37]. Additionally, the absorption peak at 845 cm$^{-1}$ or 820–825 cm$^{-1}$ suggested the presence

of sulfate groups at the C-4 position of the fucose residue or possibly associated with a low amount of substitution at the C-2 and C-3 positions in the flat-volt bond position. The absorption peak at 818 cm$^{-1}$ confirmed that most of the sulfate groups in SFP and FSFP are located at the C-2 and C-3 positions [38].

**Table 2.** Analysis of monosaccharide composition.

| Sample | Monosaccharide Composition (%) | | | | | | |
|---|---|---|---|---|---|---|---|
| | **Mannose** | **Rhamnose** | **Glucuronic Acid** | **Glucose** | **Galactose** | **Xylose** | **Fucose** |
| SFP | 2.88 | 2.51 | 0.81 | 8.79 | 16.25 | 8.25 | 60.51 |
| LP-SFP | 2.85 | 2.72 | 0.93 | 8.15 | 16.63 | 7.95 | 60.78 |
| LA-SFP | 6.07 | 1.22 | 2.14 | 10.76 | 15.62 | 6.79 | 59.86 |
| LR-SFP | 7.14 | 0.93 | 3.59 | 14.04 | 21.29 | 8.69 | 61.37 |

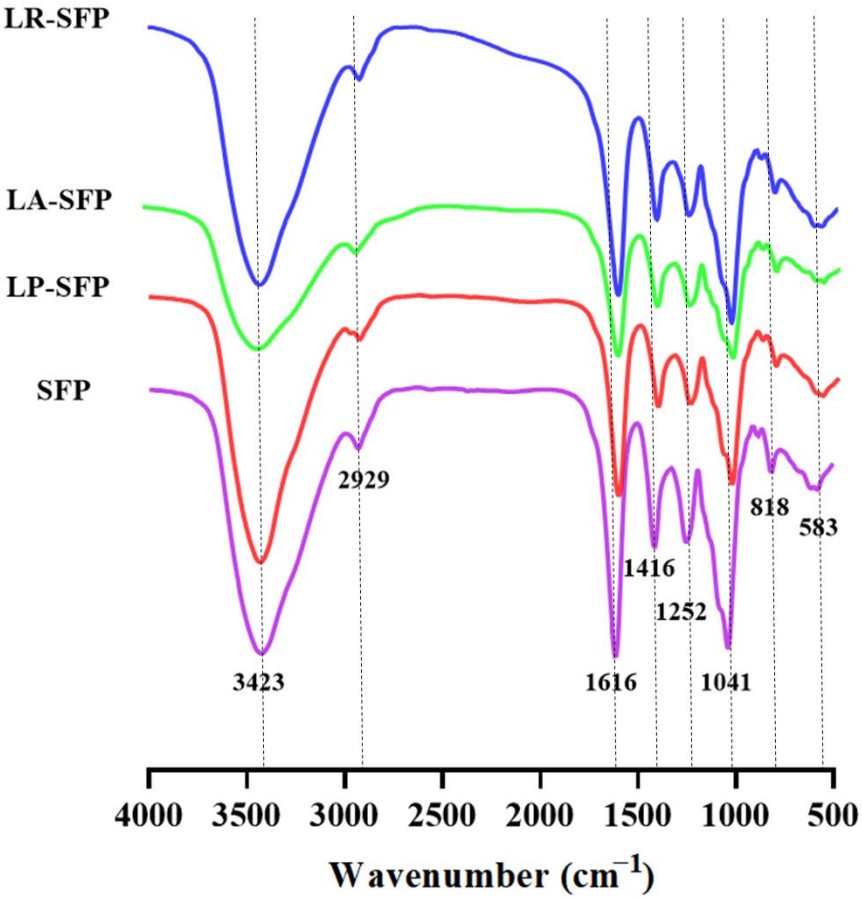

**Figure 2.** FT-IR spectra of SFP and FSFP.

*3.2. Antioxidant Activity*

3.2.1. In Vitro Antioxidant Activity

The antioxidant capacities of SFP and FSFP are shown in Figure 3. Different microbial fermentation treatments showed different antioxidant activities. As can be observed in Figure 3A, in the concentration range of 0.1–0.5 mg/mL, the ABTS radical scavenging ability increased with increasing polysaccharide concentration, showing a certain concentration–effect relationship. $EC_{50}$ values were inversely proportional to the scavenging rate, and compounds with lower $EC_{50}$ values showed higher antioxidant activity. The $EC_{50}$ values are shown in Figure 3C, and no significant ABTS radical scavenging activity was observed for fermentation-treated FSFP compared to SFP. As shown in Figure 3B, within the range

of 0.5–3 mg/mL, the DPPH radical scavenging activity exhibited an upward trend with increasing concentration and slowed down after the concentration increased to 1.5 mg/mL. As shown in Figure 3D, the $EC_{50}$ values of LA-SFP and LR-SFP were significantly lower than those of SFP, indicating that LA-SFP and LR-SFP exhibited higher antioxidant activities after fermentation.

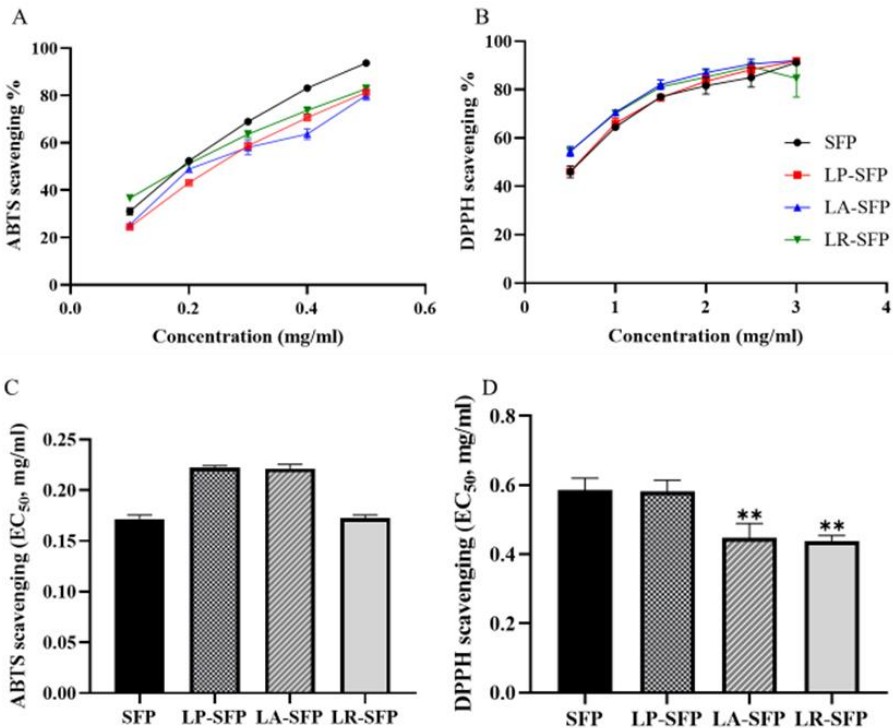

**Figure 3.** Antioxidant capacity of SFP and FSFP: (**A**) ABTS radical scavenging capacity of different concentrations of SFP and FSFP. (**B**) DPPH radical scavenging ability of SFP and FSFP at different concentrations. (**C**) ABTS radical scavenging capacity ($EC_{50}$) of SFP and FSFP. (**D**) DPPH radical scavenging capacity ($EC_{50}$) of SFP and FSFP. Results are expressed as mean $\pm$ SE; ** $p < 0.01$.

### 3.2.2. Antioxidant Activity in Zebrafish

An imbalance between antioxidant defense systems and free radical production induces ROS production, resulting in high levels of ROS, which lead to lipid, nucleic acid, and protein damage and the development of chronic diseases [39]. Therefore, the inhibition of excessive ROS production protects the organism from oxidative stress damage. The production of ROS after AAPH treatment of zebrafish embryos was studied using DCFH-DA staining (Figure 4A,B). The fluorescence intensity observed in the embryos of the AAPH-treated group was notably elevated in comparison to the untreated group. Conversely, the production of ROS in the embryos treated with SFP and FSFP exhibited a significant reduction when compared to the AAPH-treated group. These results clearly indicate that the application of polysaccharide treatment possesses a remarkable antioxidant and protective effect on zebrafish embryos. Additionally, it was observed that the antioxidant capacity of FSFP was notably superior to unfermented SFP, as demonstrated by a substantial decrease in ROS production. Notably, among the various treatments, LR-SFP exhibited the strongest capacity for scavenging ROS. As shown in Figure 4C,D, AAPH treatment significantly increased the cell mortality rate, which was significantly reduced by polysaccharide solution treatment, with a similar trend to the ROS scavenging capacity and comparable in vivo antioxidant capacity as the in vitro DPPH radical scavenging capacity. Overall, the fermented polysaccharides had stronger antioxidant activity; according to the results, LR-SFP had the strongest antioxidant activity.

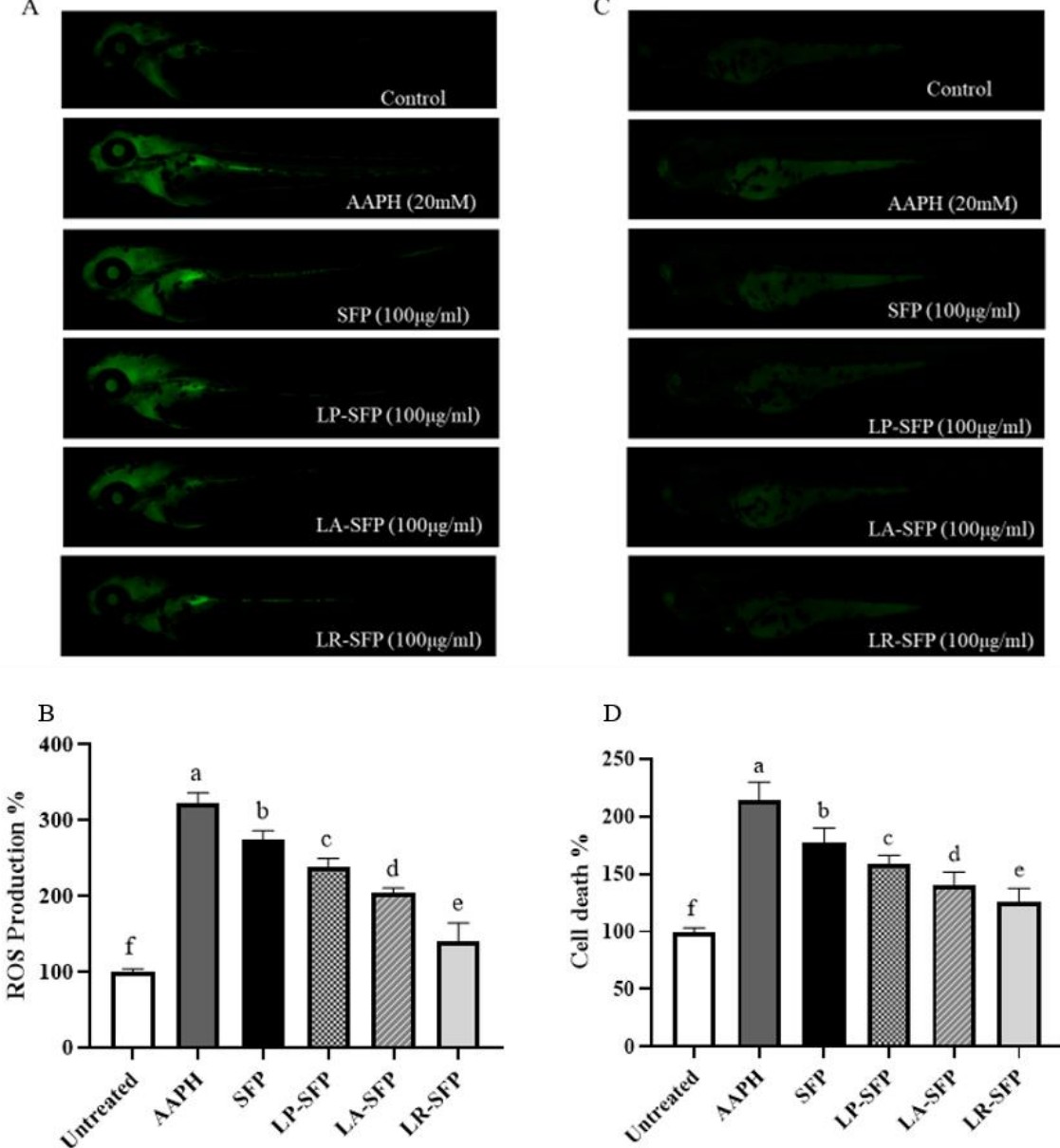

**Figure 4.** In vivo antioxidant capacity of SFP and FSFP. (**A**) AAPH-induced ROS production (DCFH-DA staining); (**C**) SFP and FSFP protected zebrafish embryos from AAPH-induced death (acridine orange staining). The quantitative results of each analysis are indicated by (**B,D**), respectively, and were analyzed by using ImageJ software. The results are presented as the mean $\pm$ SE ($n$ = 10). The use of the same letter indicates that there is no significant difference between the groups, while different letters indicate significant differences ($p < 0.05$).

In vivo enzymatic antioxidants such as SOD and CAT play a crucial role in protecting cells from oxidative damage by effectively scavenging free radicals [40]. As demonstrated in Figure 5, AAPH treatment significantly reduced the activities of SOD and CAT in zebrafish embryos when compared to the untreated group ($p < 0.05$). Figure 5A shows that polysaccharide treatment considerably enhanced SOD activity compared to the AAPH group, while the FSFP group greatly increased SOD activity compared to the SFP group. The LP-SFP- and LR-SFP-treated groups had a more significant antioxidant protective capacity. As shown in Figure 5B, zebrafish embryos treated with FSFP had stronger antioxidant capacity, among which LR-SFP had the most significant increase in CAT activity. These results indicate that fermented polysaccharides have stronger antioxidant protection.

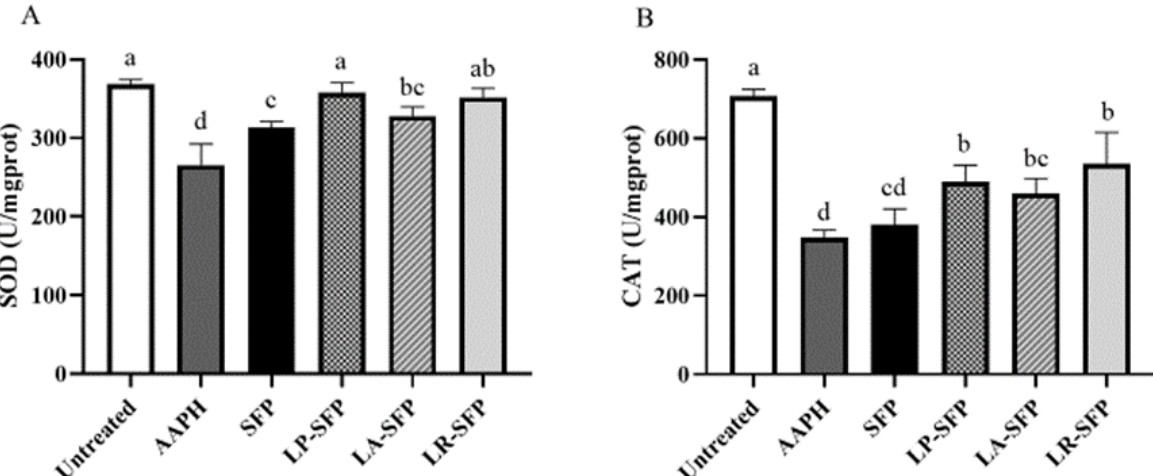

**Figure 5.** Effect of SFP and FSFP on antioxidant enzymes in AAPH-treated zebrafish embryos. (**A**) SOD activity; (**B**) CAT activity. The use of the same letter indicates that there is no significant difference between the groups, while different letters indicate significant differences ($p < 0.05$).

### 3.3. Prebiotic Activity

As shown in Figure 6, SFP and FSFP promoted the growth and proliferation of both strains. The growth of LP entered a stable phase after 12 h of incubation. Figure 6A reveals that FOS had the greatest effect on the growth rate of LP, followed by LA-SFP, and LP-SFP had the least effect on the growth rate of LP. Figure 6B shows that all five polysaccharides had effects on the growth rate of LR, among which FOS had the greatest effect on the growth rate of LR and LP-SFP had the least. After 12 h of incubation, the growth of the LR in the medium containing FOS entered the stable phase, whereas the strain in the medium containing SFP and FSFP entered the stable phase only after 36 h. This may be because the molecular weights of SFP and FSFP are higher, the fermentation rate of high-molecular-weight polysaccharides is slower than that of low-molecular-weight polysaccharides, and bacteria need a longer time to utilize the polysaccharides and provide nutrients for their growth and proliferation [41]. As shown in Figure 6C, LA-SFP and FOS had a significant effect on the proliferation of LP compared to that of SFP, whereas LP-SFP and FOS had a significant effect on the proliferation of LR (Figure 6D), and the results were consistent with the analysis in Figure 6A,B.

### 3.4. Promotes Intestinal Motility Function

Nile red is a fluorescent dye that labels the intestine; if intestinal peristalsis is increased and the lumen is drained faster, Nile red will be reduced or altogether absent in the intestine [42]. Capturing and quantifying the fluorescence intensity of the zebrafish intestine allows the assessment of the effect of polysaccharides on intestinal motility. The effects of SFP and FSFP on the intestinal motility of zebrafish are shown in Figure 7. The intestinal fluorescence intensity of zebrafish in both the SFP and FSFP groups was lower than that in the model group, indicating that both SFP and FSFP promoted intestinal motility in zebrafish. To assess the food excretion promotion rate, a calculation based on the provided Equation (6) was carried out for each treatment group. As shown in Figure 7B, the FSFP group was able to significantly increase the intestinal food excretion promotion rate of zebrafish compared to the SFP group, in which LA-SFP had stronger food excretion promotion activity.

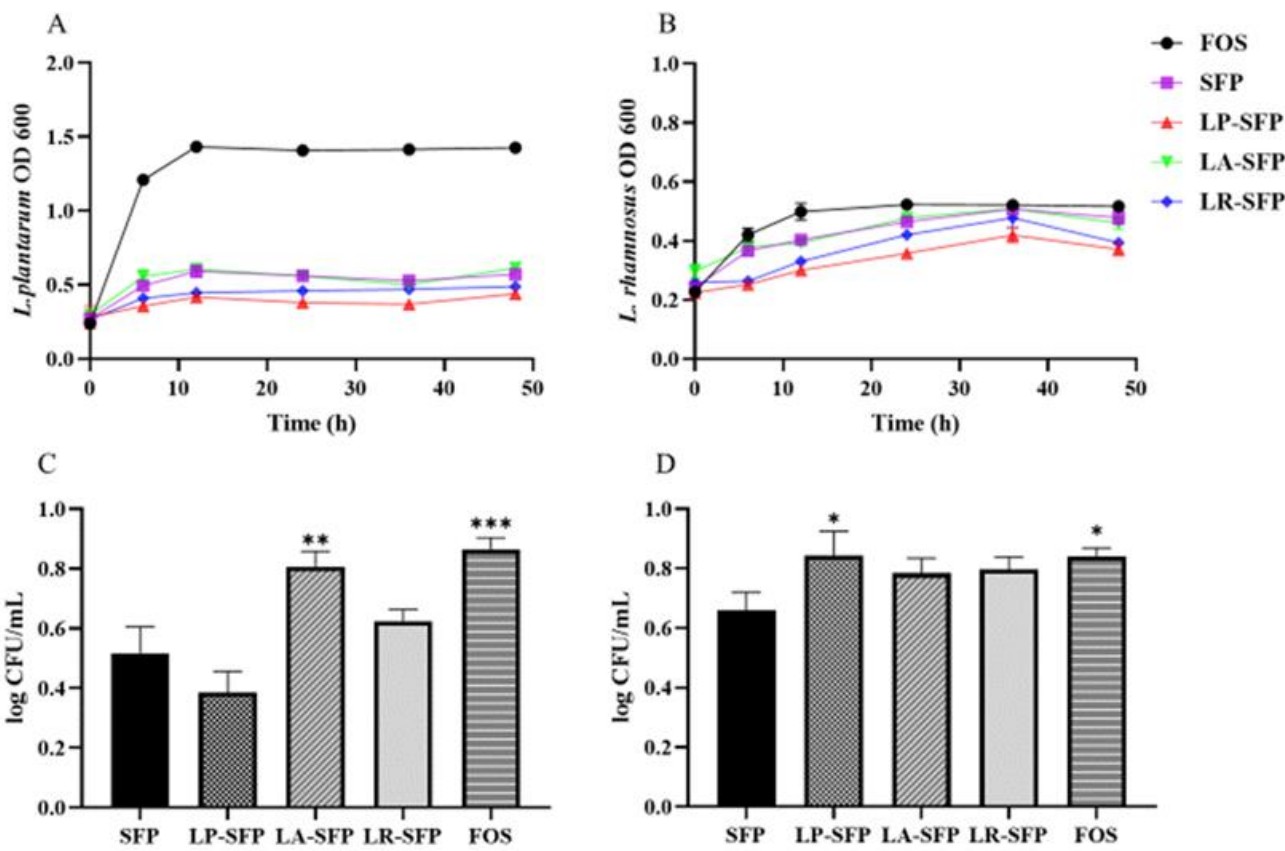

**Figure 6.** Effect of SFP and FSFP on the growth rate of *Lactobacillus plantarum* (**A**) and *Lactobacillus rhamnosus* (**B**) and the proliferation of SFP and FSFP on *Lactobacillus plantarum* (**C**) and *Lactobacillus rhamnosus* (**D**). Results are expressed as mean ± SE; * $p < 0.05$, ** $p < 0.01$, *** $p < 0.001$.

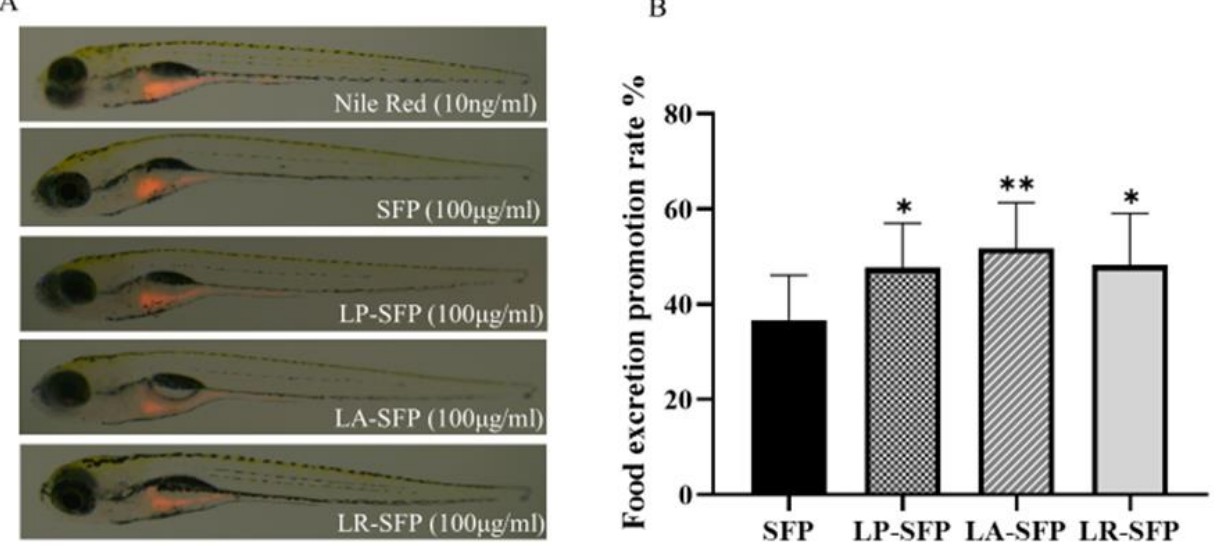

**Figure 7.** Intestinal motility promotion by SFP and FSFP in zebrafish. (**A**) Fluorogram of zebrafish intestinal staining; (**B**) quantitative analysis of food excretion facilitation. Results are expressed as mean ± SE ($n = 10$); * $p < 0.05$, ** $p < 0.01$.

## 4. Discussion

### 4.1. Effect of Fermentation on the Physicochemical Properties of SFPs

In this study, the effects of fermentation by three LABs on the physicochemical properties of SFPs were explored. LAB commonly used in food fermentation were selected to

ferment and extract polysaccharides from *S. fusiforme*. LAB can influence the composition of polysaccharides by secreting carbohydrate enzymes that also degrade plant cell walls and promote soluble carbohydrate solubilization, leading to an increase in uronic acid, fucoidan, and protein content [12,43]. Regarding the change in molecular weight, we speculate that this may be due to the degradation of the *S. fusiforme* cell wall by extracellular enzymes produced during the fermentation of *Lactobacillus* or the cleavage of the polysaccharide by binding to extracellular enzymes, resulting in a lower relative molecular weight of FSFP than SFP. Similar results have been observed where the molecular weight of LAB-fermented polysaccharides is significantly lower than that of unfermented polysaccharides [41,44]. In addition, hydrolases can also hydrolyze polysaccharides into monosaccharides or oligosaccharides, providing a carbon source for bacterial growth and metabolizing them into short-chain fatty acids [41,45]. This may account for the lower total sugar content of FSFP than that of SFP. It has been reported that the fermentation of *Lactobacillus fermentum* reduces the neutral sugar content of longan polysaccharides [46], and the fermentation of rice bran polysaccharides by *Grifola frondosa* reduces their carbohydrate content [47], similar to the results of the present study.

In this study, the fermentation of three *Lactobacillus* did not change the composition of monosaccharides but changed the proportion of monosaccharides. Related studies have shown that fermentation can change the composition and proportion of monosaccharides. Following the fermentation of wheat bran polysaccharide, the ratio of rhamnose, xylose, and arabinose increased [44]. Lily bulb polysaccharides consisted of mannose and glucose, and no mannose was detected after fermentation by LP [48]. This may be due to the abundant extracellular enzymes secreted by *Lactobacillus* [49,50]; they can alter plant tissues and enrich a class of active, multi-class ingredients in plants. These extracellular enzymes are likely responsible for altering polysaccharide content, uronic acid content, protein content, monosaccharide composition, and molecular weight.

### 4.2. Effect of Fermentation on the Activity of SFPs

### 4.2.1. Effect of Fermentation on the Antioxidant Activity of SFPs

*Lactobacillus fermentation* has been found to enhance the antioxidant activity of polysaccharides, potentially attributed to their structural characteristics [51]. Recent studies have not established the relationship between the structural characteristics of polysaccharides and their antioxidant activities, and it is challenging to infer antioxidant activity from the structure of polysaccharides. Based on the latest studies, it is inferred that increased activity may be related to monosaccharide composition, molecular weight, uronic acid, and sulfate content [52–54].

The correlation between the physicochemical properties and antioxidant activity of SFPs was previously analyzed in our laboratory. The results showed that uronic acid and glucuronide contents showed a significant linear correlation with DPPH and hydroxyl radical scavenging ability [5]. Wang et al. [55] showed that the ratio of sulfate content/fucose was a valid indicator of polysaccharide antioxidant activity, which may explain why LA-SFP and LR-SFP have better free radical scavenging abilities than SFP. The relationship between molecular weight size and antioxidant activity has been inconsistently determined in different studies. Some studies show a positive correlation between activity and molecular weight [56,57], while others show that low-molecular-weight polysaccharides have better antioxidant activity [51,58]. Liu et al. [59] showed that after ultrasonic treatment, low-molecular-weight polysaccharide extracts of *Sinopodophyllum hexandrum* fruit significantly increased SOD activity and reduced MDA content in mice; Leng et al. [60] showed that higher-molecular-weight grape polysaccharides were able to increase SOD and reduce MDA content, showing better activity. In this study, while LP-SFP and LR-SFP had the strongest ability to increase SOD activity, LP-SFP had a lower molecular weight than LR-SFP. However, the physicochemical indices also differed, and the correlation between its structure and activity was more complicated. The activity of polysaccharides is mainly influenced by their monosaccharide composition, molecular weight, glycosidic bonds,

chain conformation, uronic acid, and sulfate content [61]; however, these factors may not be dominated by a single factor but rather by a combination of several factors, and this relationship needs to be further investigated.

### 4.2.2. Effect of Fermentation on the Prebiotic Activity of SFPs

The differences in the proliferative effects of SFP and FSFP on *Lactobacillus* were related to the properties of the polysaccharides and probiotics. Because different probiotic bacteria can secrete different enzymes, leading to differences in their ability to utilize polysaccharides, LA-SFP significantly promoted the proliferation of LP, and LP-SFP significantly promoted the proliferation of LR [62]. The molecular weights of polysaccharides also affect the utilization of probiotics. In the present study, LA-SFP and LP-SFP had significantly lower molecular weights than SFP and LR-SFP and better prebiotic activity, suggesting that lower-molecular-weight polysaccharides are more easily utilized by bacteria. Kong et al. [63] showed that low-molecular-weight kelp fucoidan was more fermentable and significantly increased the levels of short-chain fatty acids and the abundance of beneficial bacteria. Huang et al. [34] showed that fermented low-molecular-weight longan pulp polysaccharides promote the proliferation of *Leuconostoc mesenteroides* and *Lactobacillus casei*. In general, within a certain molecular weight range, polysaccharides with lower molecular weights are better utilized by probiotic bacteria, thus promoting the proliferation of probiotic bacteria with better prebiotic activity [27]. In addition, the monosaccharide composition of polysaccharides may be an important factor affecting their prebiotic activity. Carbohydrates such as glucose, galactose, xylose, and fructose usually exhibit superior prebiotic activity [64]. Both SFP and FSFP contain glucose, galactose, and xylose, which may promote the growth of probiotic bacteria. In addition, because probiotic bacteria have a certain order of monosaccharide utilization [65], different molar ratios of monosaccharides may affect prebiotic activity.

The proliferation of intestinal probiotics may promote intestinal motility, and a positive correlation between intestinal motility and intestinal health has been reported [66]. *Lactobacillus* is a common beneficial intestinal microorganism that regulates the balance of the intestinal flora and ameliorates diarrhea [67]. Some studies have shown that *Lactobacillus* can improve intestinal motility in zebrafish [31]. As shown in Section 3.4, polysaccharides were able to promote intestinal motility in zebrafish, and FSFP did so more effectively than SFP, with results similar to those described in Section 3.3. FSFP had better proliferative activity of probiotics than SFP. The reason may be related to its prebiotic activity and physicochemical properties. As mentioned previously, the molecular weight and monosaccharide composition may affect prebiotic activity; however, the presence of other possible mechanisms needs to be investigated in the future.

### 5. Conclusions

In this study, SFPs were prepared through fermentation using three different *Lactobacillus* species, whose physical and chemical properties were characterized, and the activities of the SFP and FSFP were evaluated using in vitro and in vivo models. The results showed that the total soluble sugars, sulfate, uronic acid, fucose, and molecular weights of polysaccharides extracted by the fermentation of different strains changed, and their biological activities were also different. Compared to SFP, LA-SFP and LR-SFP had a more significant DPPH scavenging ability; LR-SFP had the most significant ROS and cell mortality inhibitory activity; LA-SFP significantly promoted LP growth; LP-SFP significantly promoted LR proliferation; and LA-SFP had the most significant food-excretion-promoting activity. Therefore, when SFP with specific activities is required, fermentation strains that can enhance these specific activities should be selected. In conclusion, *Lactobacillus* fermentation is an effective method for biomodifying polysaccharides, and FSFP has the potential to be developed as a functional food candidate with stronger antioxidant, prebiotic, and pro-intestinal motility activities than SFP.

**Author Contributions:** Conceptualization, L.S. and M.W.; investigation, D.O., J.S., C.C. and C.Y.; writing—original draft preparation, Y.Y.; writing—review and editing, Y.Y. and L.S.; supervision, L.S.; project administration, L.S.; funding acquisition, L.S. and M.W. All authors have read and agreed to the published version of the manuscript.

**Funding:** This study was financially supported by the Agricultural national standards and industry standards revision project (NYB-22270), Wenzhou Major Science and Technology Innovation Tackling Project (ZN2022009), and Wenzhou Science and Technology Specialists Project (X20210015) as part of the first batch of the list of hanging projects of Wenzhou prefabricated vegetable industry research and development being unveiled.

**Data Availability Statement:** The authors confirm that the data supporting the findings of this study are available within the article.

**Conflicts of Interest:** The authors declare no conflict of interest.

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
