# Peer review of "Physicochemical Properties and Biological Characteristics of Sargassum fusiforme Polysaccharides Prepared through Fermentation of Lactobacillus"

_fermentation, doi:10.3390/fermentation9090835_

Round 1
Reviewer 1 Report
The manuscript entitled “Physicochemical properties and biological characteristics of Sargassum fusiforme polysaccharides prepared by fermentation of Lactobacillus” reported the physicochemical properties and biological activities of Sargassum fusiforme polysaccharides (SFPs) obtained by fermentation of three species of Lactobacillus. The polysaccharide obtained without fermentation was also studied.
I recommend that this manuscript should be accepted for publication.

Author Response
Thank you for reviewing our manuscript and for your kind words about it.
Reviewer 2 Report
This paper talks about how Sargassum fusiforme polysaccharides were changed by fermentation with different Lactobacillus species and how their physical, chemical, and biological properties were measured. This paper can be published in this journal after suggested revisions:
- The purification of polysaccharides should be carried out.
- What are the ingredients in the bacterial broth?
- Is the fermentation carried out under static or shaking conditions?
- Is it possible for the authors to run NMR on the dried samples?
- In the discussion section, the author said that: "In addition, fermentation can also change the monosaccharide composition". Proof is needed, and what type of change in the chemical structure of the polysaccharide.
- Some grammatical mistakes were observed.
- Some grammatical mistakes were observed.
Author Response
Point 1:
The purification of polysaccharides should be carried out.
Response 1:
Thank you for your comments. The focus of this study is to compare the effects of different strains fermentation on the physical and chemical properties and activity of Sargassum fusiforme polysaccharides. In the future, we will choose fermentation strains that can significantly improve the functional activity, optimize the conditions of fermentation to prepare polysaccharides, and purify and identify the structure of polysaccharides.
Point 2:
What are the ingredients in the bacterial broth?
Response 2:
Bacterial broth refers to the MRS broth medium of activated strains. The main components include casein peptone, beef extract, yeast powder, glucose, sodium acetate, citric acid diamine, Tween, dipotassium hydrogen phosphate, magnesium sulfate and magnesium sulfate.
Point 3:
Is the fermentation carried out under static or shaking conditions?
Response 3:
The fermentation is carried out under static conditions. We have added this condition in 2.3.
Point 4:
Is it possible for the authors to run NMR on the dried samples?
Response 4:
Similar to point 1, the focus of this study is to compare the effects of different strains fermentation on the physicochemical properties and functional activities of polysaccharides. In the next work, we intend to screen the best strain fermentation to prepare Sargassum fusiforme polysaccharides, and analyze the polysaccharides by MS and NMR to identify the polysaccharide structure and further study the functional activity mechanism.
Point 5:
In the discussion section, the author said that: "In addition, fermentation can also change the monosaccharide composition". Proof is needed, and what type of change in the chemical structure of the polysaccharide.
Response 5:
Thank you for your comment, this sentence in the discussion may not be appropriate, the literature reported that fermentation can change the monosaccharide composition and monosaccharide ratio, in this study we used PMP derivatization method to study the monosaccharide composition of polysaccharides, and found that fermentation of the three strains we used did not change the monosaccharide composition of the polysaccharides but it could change the ratio of the monosaccharides. We have corrected this in the discussion section of the text:
“In this study, the fermentation of three Lactobacillus did not change the composition of monosaccharides, but changed the proportion of monosaccharides. Related studies have shown that fermentation can change the composition and proportion of monosaccharides. Following fermentation of wheat bran polysaccharide, the ratio of rhamnose, xylose, and arabinose increased [43]. Lily bulb polysaccharides consisted of mannose and glucose, and no mannose was detected after fermentation by LP [47]. This may be due to the abundant extracellular enzymes secreted by Lactobacillus. [48, 49], they can alter plant tissues and enrich a class of active, multi-class ingredients in plants. These extracellular enzymes are likely responsible for altering polysaccharide content, uronic acid content, protein content, monosaccharide composition, and molecular weight.”
Reviewer 3 Report
Physicochemical properties and biological characteristics of Sargassum fusiforme polysaccharides prepared by fermentation of Lactobacillus
Main comment: The Sargassum fusiform extraction methodology is not clearly presented. It is needed to clarify which was the salt content and final pH before ethanol precipitation. Polysaccharides extraction depend on these conditions. Neither the literature nor the authors' introduction are consistent with the results shown in Table 1. Information regarding alginate concentration is not provided because it is absent from the methodology and the outcomes. This should be referred as one of the primary polysaccharides found in Sargassum fusiforme is alginate. What does, for instance, the column "total sugar" in Table 1 mean? The monosaccharides identified by chromatography, or does it refer to all carbohydrates identified by the phenol sulfuric acid method, including mono-, di-, oligo-, and polysaccharides?
In line 198, the authors refer to soluble sugars. Depending on the extraction methodology, the extracted alginate precipitated with ethanol could be in a water insoluble form.
In my opinion, authors should reformulate the methodology description and the results presentation. They should identify unequivocally what they have quantified, and discuss the results by comparing them with the literature.
Other comments:
Comment 1: The authors should indicate for the first time the meaning of all abbreviations. It could be dubious for some of them, which is the meaning. And it is not correct for the reader to understand the meaning only from the context.
For example, for SOD, write for the first time: superoxide dismutase (SOD) activity assay, etc.
Comment 2: The authors should make clear the following steps:
1. What is the strain broth final concentration after adding to the 30g algae material-liquid ratio of 1:10 mixture?
“2.3. Preparation of fermented SFPs …. Bacterial broth containing 4% (v/v) LP, LA, or LR was added and fermented at 37 °C for 24 h. “
Comment 3: The authors should indicate the final pH in the mixture of bacterial broth and algae material-liquid! Because some acidic polysaccharides, or high MW polysaccharides are not easily dissolved in hot water and depend on the medium pH to be extracted!
Comment 4: The authors should indicate the analytical method used for the monosaccharide composition determination. Was it HPLC? Which equipment and detector?
2.4.1. Chemical characteristics
Authors should introduce these corrections:
Line 29, eliminate the ref. 1 in the sub-title “Introduction”. It is not normal and it is not clear why it is there
Line 51, low capital in “Probiotic-diet “
Line 72, D-Glucose and not D-glucose
Line 103 and 109: Indicate the column particle size
Line 104, pH and not PH
Lines 153,154, correct to a full sentence and use the same type of writing as in line 127-129: “Where A is the number of viable bacteria at 0 h of incubation (CFU/mL) and B the number of viable bacteria after 48 h of incubation (CFU/mL). “
Equation 1: eliminate % from “100%”
Equation 4: eliminate % from “100%” and add the concentration unit before the equal sign
Line 191: “All experiments were repeated at least three times, ...”. The authors should indicate exactly how many times each experiment has been replicated. “at least” has no meaning.
Author Response
Point 1:
The Sargassum fusiform extraction methodology is not clearly presented. It is needed to clarify which was the salt content and final pH before ethanol precipitation. Polysaccharides extraction depend on these conditions. Neither the literature nor the authors' introduction are consistent with the results shown in Table 1. Information regarding alginate concentration is not provided because it is absent from the methodology and the outcomes. This should be referred as one of the primary polysaccharides found in Sargassum fusiforme is alginate. What does, for instance, the column "total sugar" in Table 1 mean? The monosaccharides identified by chromatography, or does it refer to all carbohydrates identified by the phenol sulfuric acid method, including mono-, di-, oligo-, and polysaccharides?
Response 1:
Thank you for your valuable comments, we have added information about the extraction conditions at 2.3 in the text: “S. fusiforme was washed, dried, and crushed. Before fermentation, 30 g of algae powder was taken, and water was added in a material-liquid ratio of 1:10, adjust the pH to 6.8±0.1, followed by sterilization for 20 min at 121 ℃. Bacterial broth containing 4% (v/v) LP, LA, or LR was added to a final concentration of 6.5 log CFU/mL, and static fermentation was carried out at 37°C for 24 h.. A sterile water control was used instead of bacterial broth. The fermentation reaction for both the fermented (FSFP) and unfermented (SFP) groups was stopped by heating at 100 °C for 10 min, the supernatant was collected by centrifugation, and the final pH of the fermentation broth was determined to be 4.7 ± 0.1. The supernatant was concentrated under reduced pressure, and 3 times the volume of anhydrous ethanol was added. The resulting precipitate was obtained by centrifugation at 4 °C for 10 h. Fermentation of S. fusiforme with LR, LA, and LR resulted in fermented SFP (FSFP: LP-SFP, LA-SFP, and LR-SFP, respectively), while unfermented SFP was obtained through water extraction.”
Regarding the determination of alginate content, the relevant assay method has been added at 2.4.1 in the text and the results of alginate concentration for each polysaccharide are presented in Table 1. In short, accurately weigh 1 g of the sample, add 2 mol/L hydrochloric acid 30 mL to soak for 12 h, filter, wash the filter residue to the filtrate, add silver nitrate solution to the filtrate without white precipitation, filter residue with 30 mL 0.1 mol/L calcium acetate solution, soak for 2 h, add 50 mL double distilled water to mix evenly, use phenolphthalein as an indicator, and titrate with sodium hydroxide standard solution. Alginate content was calculated according to the following formula:
X (%)= (((V-V0) × C × 0.2160)/ m) × 100 (1)
where V is the volume of sodium hydroxide standard solution consumed by the titration sample (mL), V0 is the volume of sodium hydroxide standard solution consumed by the titration blank sample (mL), C is the concentration of sodium hydroxide standard solution (mol/L), 0.2160 is the mass of sodium alginate comparable to 1.00 mL of sodium hydroxide standard titration solution [c(NaOH)=1.000 mol/L] ( g), m is the sample mass (g), and 100 is the unit conversion factor. “
The total sugar in this paper refers to all carbohydrates determined by phenol sulfuric acid method.
Point 2:
In line 198, the authors refer to soluble sugars. Depending on the extraction methodology, the extracted alginate precipitated with ethanol could be in a water insoluble form.
Response 2:
Thank you for pointing out the problem, we have made changes in the text.
Point 3:
The authors should indicate for the first time the meaning of all abbreviations. It could be dubious for some of them, which is the meaning. And it is not correct for the reader to understand the meaning only from the context. For example, for SOD, write for the first time: superoxide dismutase (SOD) activity assay, etc.
Response 3:
We have made modifications in the text according to your suggestions.
Point 4:
The authors should make clear the following steps:
- What is the strain broth final concentration after adding to the 30g algae material-liquid ratio of 1:10 mixture? “2.3. Preparation of fermented SFPs …. Bacterial broth containing 4% (v/v) LP, LA, or LR was added and fermented at 37 °C for 24 h. “
Response 4:
Thank you for your comments, same as point 1 we have added the information at 2.3.
Point 5:
The authors should indicate the final pH in the mixture of bacterial broth and algae material-liquid! Because some acidic polysaccharides, or high MW polysaccharides are not easily dissolved in hot water and depend on the medium pH to be extracted!
Response 5:
Relevant information has been added at 2.3.
Point 6:
The authors should indicate the analytical method used for the monosaccharide composition determination. Was it HPLC? Which equipment and detector?
Response 6:
We have added information on methods, equipment and detectors at 2.3 in the text.
“The monosaccharide composition of FSFP and SFP was determined using the PMP (1-phenyl-3-methyl-5-pyrazolinone) derivatization method [22]. A Waters XBridge C18 (4.6 mm ×250 mm, 5μm) column was used with a 1260 Infinity II high performance liquid chromatography (HPLC) system (Agilent) equipped with a DAD detector. Mobile phase A was phosphate buffer solution (Na2HPO4-NaH2PO4) (PBS 0.1 mol/L pH=6.9), mobile phase B was acetonitrile, phase A: B=83:17, flow rate was 1 mL/min, injection volume was 20 μL, detection wavelength was 254 nm, and the column temperature was 25 ℃.“
Point 7:
Authors should introduce these corrections:
Line 29, eliminate the ref. 1 in the sub-title “Introduction”. It is not normal and it is not clear why it is there
Line 51, low capital in “Probiotic-diet “
Line 72, D-Glucose and not D-glucose
Line 103 and 109: Indicate the column particle size
Line 104, pH and not PH
Lines 153,154, correct to a full sentence and use the same type of writing as in line 127-129: “Where A is the number of viable bacteria at 0 h of incubation (CFU/mL) and B the number of viable bacteria after 48 h of incubation (CFU/mL). “
Equation 1: eliminate % from “100%”
Equation 4: eliminate % from “100%” and add the concentration unit before the equal sign
Line 191: “All experiments were repeated at least three times, ...”. The authors should indicate exactly how many times each experiment has been replicated. “at least” has no meaning.
Response 7:
Thank you for your comments, we have been modified in the text.

Round 2
Reviewer 3 Report
The authors response to the comments and the proposed corrections are complete and satisfactory.
Thank you.